# The prognostic impact of lymph node dissection for clinically node-negative upper urinary tract urothelial carcinoma in patients who are treated with radical nephroureterectomy

**Hsiang-Chen Hsieh**[1], **Chun-Li Wang**[2,3], **Chuan-Shu Chen**[1,2], **Cheng-Kuang Yang**[1], **Jian-Ri Li**[1,2,4], **Shian-Shiang Wang**[1,2,4], **Chen-Li Cheng**[1,2], **Chia-Yen Lin**[1,2]*, **Kun-Yuan Chiu**[1,5]*

1 Division of Urology, Department of Surgery, Taichung Veterans General Hospital, Taichung, Taiwan, 2 Institute of Medicine, Chung Shan Medical University, Taichung, Taiwan, 3 Department of Family Medicine, Taichung Veterans General Hospital, Taichung, Taiwan, 4 Department of Medicine and Nursing, Hungkuang University, Taichung, Taiwan, 5 Department of Applied Chemistry, National Chi Nan University, Nantou, Taiwan

* lcyhank.tw@gmail.com (CYL); chiu37782002@yahoo.com (KYC)

## Abstract

### Background

To evaluate the prognostic impact of lymph node dissection (LND) in patients who underwent radical nephroureterectomy (RNU) with bladder cuff excision (BCE) for clinically node-negative (cN0) upper urinary tract urothelial carcinoma (UTUC).

### Methods

We retrospectively enrolled 520 patients with cN0 UTUC in a single tertiary referral center from 2000 to 2015. The patients were divided into three groups: patients with and without pathologically proved lymph node metastasis (pN1–3 and pN0, respectively) and patients without LND (pNx). We analyzed associations between overall survival (OS)/ disease-free survival (DFS)/ cancer-specific survival (CSS) and clinical characteristics.

### Results

The patients were divided into three groups (pN1–3, pN0 and pNx with 20, 303, and 197 patients, respectively). OS/DFS/CSS in the pN1–3 group were significantly worse (all p<0.001) compared with the pN0 group. However, there were no significant differences between the pNx and pN0 groups. In the multivariate analyses, CSS was only affected by age [(hazard ratio (HR) = 1.03, p = 0.008]), positive surgical margin (HR = 3.38, p<0.001) and pathological T3–4 stages (HR = 4.07, p<0.001). In the subgroup analyses for patients with LND, locally advanced disease (pT3 and pT4) had significantly more metastases [T3–4: 13.91% (16/115) vs. T0–2: 1.92% (4/208), p<0.001].

**Data Availability Statement:** All relevant data are within the paper and its Supporting Information files.

**Funding:** The authors received no specific funding for this work.

**Competing interests:** The authors have declared that no competing interests exist.

## Conclusions

In the pN0 group, LND for cN0 UTUC did not show therapeutic benefits in terms of DFS, CSS, and OS. However, LND with RNU allowed optimal tumor staging, through patients still had a poor prognosis. Clinically occult LN metastases were found in 6.2% of our patients.

## Introduction

Upper urinary tract urothelial carcinoma (UTUC), which originates from the pyelocaliceal cavities and ureter, is an infrequent malignancy, only accounting for 5–10% of urothelial carcinomas (UCs) [1–3].

Taiwan is the one of the endemic area where UTUC accounts for approximately a third of all urothelial tumors. According to the Taiwan Cancer Registry Annual Report, the age-standardized incidence rate of UTUC is 3.71 and 3.99 per 100,000 population in men and women, respectively [4, 5].

Radical nephroureterectomy (RNU) and ipsilateral bladder cuff excision (BCE) with or without lymph node dissection (LND) is the standard surgical intervention for localized UTUC [1, 6]. However, due to the poor prognostic nature with a high risk of lymphatic spread and disease progression [7–9], the five year survival for patients with UTUC is <50% and <10% for stage pT2/3 and pT4 disease, respectively [3, 10–12].

Regarding the management of UC of the urinary bladder (UCUB), the National Comprehensive Cancer Network (NCCN) Practice Guidelines in Oncology has suggested the standard therapy of neo-adjuvant chemotherapy (NAC) followed by radical RNU with LND for stages ≥cT2 [7, 13]. Recent systematic reviews have also reported on the survival benefit of NAC in patients with locally advanced UTUC [14–17].

In addition, adjuvant chemotherapy (AC) should be considered for patients with pT3–4 or nodal-positive disease. A phase III POUT trial has demonstrated the benefit of platinum-based AC for patients with locally advanced UTUC [18]. A meta-analysis has reported that platinum-based AC is associated with improved disease-free survival (DFS) for locally advanced UCs [19]. Alternatively, immune checkpoint inhibitors (ICIs) that had been investigated as an adjuvant treatment of UTUC in the CheckMate-274 trial (nivolumab) may be considered [20]. Alessandro et al. has suggested receiving ICIs have survival benefits in programmed cell death ligand 1 (PD–L1) for patients with positive metastatic UC (mUC) [21].

Regarding LND at the time of RNU, therapeutic benefits have been reported for patients with UTUC, particularly those with muscle-invasive or locally advanced disease [22, 23]. According to the NCCN guidelines, LND should be performed in patients with high-grade disease, large tumors, and tumors invading the renal parenchyma [13]. However, the benefits of LND for patients with cN0 disease remain debatable, and the procedure is not standardized.

In this study, we aimed to evaluate the prognostic impact of LND on overall survival (OS), DFS, and cancer-specific survival (CSS) in patients undergoing RNU with BCE for cN0 UTUC.

## Materials and methods

We reviewed 728 patients with UTUC who received RNU with ipsilateral BCE between 2001 and 2015 in a retrospectively built UTUC database at the Taichung Veterans General Hospital, a single tertiary referral center in central Taiwan. We excluded patients who had NAC,

previous or concurrent cystectomy, incomplete clinical data (without clinical status), distant metastasis, clinical lymph node involvement ($\geq$cN1), and short follow-up duration (<one year). Finally, we identified 520 patients, whom we then divided into three groups: those with pathologically confirmed lymph node metastases (pN1–3), pN0, and without LND (pNx).

Surgeries were performed by seven well-experienced urological surgeons in our hospital. Until 2007, we performed hilar and regionalLND only in patients with clinically/surgically suspicious lymph node metastasis. Beginning in 2008, at least hilar LND with or without regional LND was routinely performed with RNU. Hilar LND was performed over the renal vein root at the right side and renal artery root at the left side. The templates of regional LND depended on the tumor location, in that para-aortic and peri-caval LND were performed for renal pelvic or proximal ureteral tumors and pelvic LND, for middle or distal ureter tumors. AC was considered for patients with advanced tumor features. The regimens of chemotherapy were based on cisplatin or carboplatin and depended on renal function. However, the indications of performing AC or regional LND were based on the patient's clinical stage and the surgeons' preference.

The Institutional Review Board (IRB) of Taichung Veteran General Hospital approved the current study, and informed written consent was obtained from all of the participants (IRB No. CE13240A-3). The procedures performed were in accordance with the Declaration of Helsinki guidelines.

The endpoints of this study after RNU were OS, DFS, and CSS. DFS was defined as local recurrence and lymph node and/or distant metastasis, not including recurrences at the contralateral upper urinary tract or bladder. The time duration from the date of treatment for UTUC was defined as OS. CSS was the time duration from the date of diagnosis to death solely due to UTUC. In addition, we performed a subgroup analysis on patients with LND (pN0 and pN+), with 278 patients in the hilar-only and 45 patients in the regional LND groups.

Correlations between the three groups and other clinic-pathological characteristics were tested using the chi-square or Kruskal-Wallis test. The survival curve for the presence of LND (patients with/without lymph node metastasis) was estimated via the Kaplan-Meier method, and differences were assessed using the log-rank statistic (Mantel-Cox). Univariate and multivariate analyses were performed with Cox proportional hazards regression models to determine the impacts of LND on OS and CSS. Results were showed with hazard ratios (HRs) to reflect relative risks at 95% confidence intervals (CIs). All reported p-values were two-sided, and statistical significance was set at p$\leq$0.05.

## Results

A total of 520 patients were included in our study. They were divided into three groups based on histopathology: 303 (58.3%) with pN0 disease, 20 (3.8%) with pN+ disease (pN1-3), and 197 (37.9%) with no LND (pNx). The mean follow-up duration was 47.63 months [standard deviation (SD) = 28.96]. The respective median ages of the three groups at diagnosis were: (a) 68.3 years [interquartile range (IQR) = 62.1–76.6], (b) 70.2 years (IQR = 57.5–78.5), and (c) 67.5 years (IQR = 57.8–75.7). There was a significant difference, respectively, in the rate of advanced pathological stage ($\geq$T2; 46.8%, 85.0%, and 43.8%, p<0.001), tumor grade 3 (68.6%, 100.0%, and 43.7%, p<0.001), positive lymphovascular invasion (17.5%, 75.0%, and 15.7%, p<0.001), positive surgical margin (8.6%, 40.0%, and 2.1%, p<0.001) and post-AC (18.8%, 70.0%, and 18.3%, p <0.001) for the three groups (Table 1).

During the following-up, 152 patients (29.2%) experienced disease recurrence, 79 (15.2%) died of UTUC, and 61 (11.7%) died of other causes. The two-year OSs were 85.9%, 50.0%, and 88.6%, respectively, in the PN0, PN+, and PNx groups. We found that the pN+ group,

**Table 1. Association of LND status and clinic-pathological characteristics of patients undergoing RNU with BCE for cN0 UTUC.**

| | PN0 (n = 303) | PN1-N3 (n = 20) | PNx (n = 197) | *P* value |
|---|---|---|---|---|
| **Gender** | | | | 0.026* |
| Male | 111 (36.6%) | 9 (45.0%) | 96 (48.7%) | |
| Female | 192 (63.4%) | 11 (55.0%) | 101 (51.3%) | |
| **Age, years** | 68.3 (62.1–76.6) | 70.6 (57.5–78.5) | 67.6 (57.8–75.7) | 0.448 |
| **BMI** | 23.6 (21.3–25.6) | 24.4 (22.3–26.5) | 24.0 (21.9–26.3) | 0.256 |
| **Performance Status ECOG** | | | | **<0.001**** |
| 0 | 50 (16.5%) | 5 (25.0%) | 2 (1.0%) | |
| 1 | 187 (61.7%) | 11 (55.0%) | 173 (87.8%) | |
| 2 | 64 (21.1%) | 3 (15.0%) | 19 (9.6%) | |
| 3 | 2 (0.7%) | 1 (5.0%) | 2 (1.0%) | |
| 4 | 0 (0.0%) | 0 (0.0%) | 1 (0.5%) | |
| **Smoking status** | | | | 0.639 |
| Never | 215 (76.0%) | 15 (78.9%) | 137 (70.6%) | |
| Current | 39 (13.8%) | 2 (10.5%) | 29 (14.9%) | |
| Former | 29 (10.2%) | 2 (10.5%) | 28 (14.4%) | |
| **Comorbidity** | | | | |
| CAD/HTN | 187 (61.7%) | 11 (55.0%) | 109 (55.3%) | 0.341 |
| DM | 65 (21.5%) | 5 (25.0%) | 37 (18.8%) | 0.680 |
| COPD/Asthema | 6 (2.0%) | 0 (0.0%) | 9 (4.6%) | 0.176 |
| CVA | 11 (3.6%) | 1 (5.0%) | 9 (4.6%) | 0.852 |
| CKD(Cr>1.5, non-uremic status) | 55 (18.2%) | 3 (15.0%) | 71 (36.0%) | **<0.001**** |
| Uremia at diagnosis | 44 (14.5%) | 2 (10.0%) | 23 (11.7%) | 0.596 |
| HBV or HCV carrier | 26 (8.6%) | 1 (5.0%) | 35 (17.8%) | **0.005**** |
| **Pathological T** | | | | **<0.001**** |
| T0 | 1 (0.3%) | 0 (0.0%) | 0 (0.0%) | |
| T1 | 160 (52.8%) | 3 (15.0%) | 111 (56.3%) | |
| T2 | 43 (14.2%) | 1 (5.0%) | 33 (16.8%) | |
| T3 | 91 (30.0%) | 8 (40.0%) | 46 (23.4%) | |
| T4 | 8 (2.6%) | 8 (40.0%) | 7 (3.6%) | |
| **Multifocalty** | | | | 0.065 |
| No | 182 (60.1%) | 14 (70.0%) | 138 (70.1%) | |
| Yes | 121 (39.9%) | 6 (30.0%) | 59 (29.9%) | |
| **Tumor cell differentiation** | | | | |
| CIS | | | | **0.020*** |
| Negative | 246 (81.2%) | 17 (85.0%) | 178 (90.4%) | |
| Positive | 57 (18.8%) | 3 (15.0%) | 19 (9.6%) | |
| Tumor grading | | | | **<0.001**** |
| G1 | 7 (2.3%) | 0 (0.0%) | 11 (5.6%) | |
| G2 | 88 (29.0%) | 0 (0.0%) | 100 (50.8%) | |
| G3 | 208 (68.6%) | 20 (100.0%) | 86 (43.7%) | |
| **Lymphovascular invasion** | | | | **<0.001**** |
| Negative | 249 (82.5%) | 5 (25.0%) | 161 (84.3%) | |
| Positive | 53 (17.5%) | 15 (75.0%) | 30 (15.7%) | |
| **Surgical margin** | | | | **<0.001**** |
| Negative | 276 (91.4%) | 12 (60.0%) | 187 (97.9%) | |
| Positive | 26 (8.6%) | 8 (40.0%) | 4 (2.1%) | |
| **Adjuvant chemotherapy** | | | | **<0.001**** |

(*Continued*)

**Table 1.** (Continued)

|  | PN0 (n = 303) | PN1-N3 (n = 20) | PNx (n = 197) | *P* value |
|---|---|---|---|---|
| No | 246 (81.2%) | 6 (30.0%) | 161 (81.7%) | |
| Yes | 57 (18.8%) | 14 (70.0%) | 36 (18.3%) | |

Chi-square test. Kruskal-Wallis test, Median (IQR).

*\*P<0.05*

*\*\*P<0.01.*

ECOG = Eastern Cooperative Oncology Group. CAD/HTN = coronary artery disease / hypertension. DM = diabetes mellitus. COPD = chronic obstructive pulmonary disease. CVA = cerebrovascular accident. CKD = chronic kidney disease. HBV = hepatitis B virus. HCV = hepatitis C virus. CIS = carcinoma in situ. G = Grade.

compared with the pN0 group has a significantly worse OS (five years, 76.6% vs. 25.9%, p<0.001), DFS (75.8% vs. 29.2%, p<0.001), and CSS (85.0% vs. 25.9%, p<0.001). On the other hand, no significant difference was found in the pNx group compared with the pN0 group in terms of DFS, CSS and OS, through there may have been a worse trend in the pNx group compared to the pN0 group in the five-year OS and DFS (64.0% vs. 76.6%, P = 0.101 and 64.5% vs. 75.8%, p = 0.204) (Fig 1) (S1–S3Tables).

In the univariate analysis, worse CSS was found to be correlated with age (HR = 1.03, p = 0.011), smoking status, patients with pN1–3 (HR = 6.93, p<0.001), pathological T3–4 stage (HR = 6.17, p<0.001), positive lymphovascular invasion (HR = 4.46, p<0.001), positive surgical margin (HR = 8.49, p<0.001), and post-AC (HR = 2.09, p = 0.002). However, in the multivariate analysis, only age (HR = 1.03, p = 0.017), patients with pN1–3 (HR = 2.10, p = 0.049), pathological T3–4 stage (HR = 4.42, p< 0.001), and positive surgical margin (HR = 3.37, p<0.001) significantly affected CSS (Table 2).

In the subgroup analysis for patients with LND (pN0 and pN+), 20 patients were LN-positive (6.2% of 323 patients). In the N+ group, locally advanced disease (pT3 and pT4) had a significantly higher rate of node metastasis [T3–4 vs. T0–2: 13.91% (16/115) and 1.92% (4/208), p<0.001]. We also found a trend in the N+ group had more grade 3 tumors [68.6% (208/303) and 100.0% (20/20)], more instances of lymphovascular invasion [17.5% (53/303) and 75.0% (15/20)], and a higher margin positive rate [8.65 (26/303) and 40.0% (8/20)]. In addition, 278 receiving RNU had hilar-only LND, and 45 had regional LND. The average number of removed lymph nodes were 1.0 (range 0.0 to 5.0) in hilar-only LND and 11.0 (range 6.0 to 41.0) in regional LND. Between these two groups of patients, the regional LND group had more dissected nodes [1.0 (0.0–5.0) vs. 11.0 (6.0–41.0), p<0.001], and more node metastases

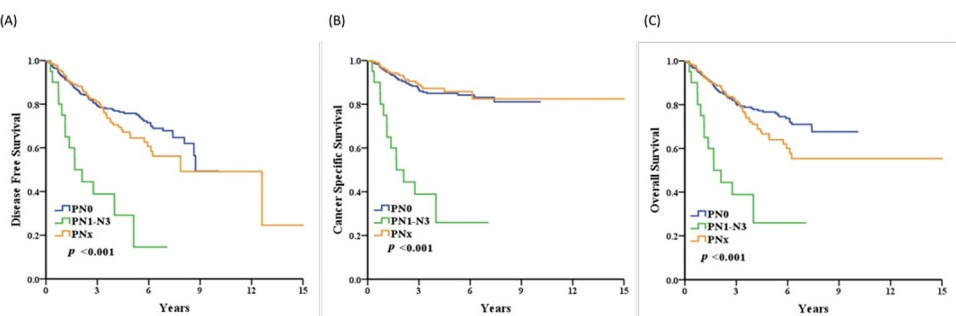

**Fig 1.** Kaplan-Meier curves of DFS (A), CSS (B) and OS (C) for 520 patients with pathologically proved lymph node status (pN1-3 and pN0) or without lymph node dissection (pNx)in clinical node-negative upper urinary tract urothelial carcinoma undergoing radical nephroureterectomy.

**Table 2. Univariate and multivariate Cox regression analysis predicting CSS, OS, and DFS for 520 cN0 UTUC patients with pathologically proved lymph node status (pN1–3 and pN0) or without LND (pNx) undergoing RNU with BCE.**

| | CSS | | | | | |
|---|---|---|---|---|---|---|
| | Univariate | | | Multivariable | | |
| | Hazard ratio | 95%CI | *p*value | Hazard ratio | 95%CI | *P*value |
| **Gender** | | | | | | |
| Male | Reference | Reference | | | | |
| Female | 0.64 | (0.41–1.00) | 0.050 | | | |
| **Age, years** | 1.03 | (1.01–1.05) | **0.011**[*] | 1.03 | (1.01–1.05) | **0.017**[**] |
| **BMI** | 0.94 | (0.88–1.01) | 0.081 | | | |
| **Group** | | | | | | |
| PN0 | Reference | Reference | | Reference | Reference | |
| PN1–3 | 6.93 | (3.71–12.97) | **<0.001**[**] | 2.01 | (1.00–4.39) | 0.049[*] |
| PNx | 0.88 | (0.53–1.47) | 0.636 | 0.97 | (0.56–1.69) | 0.915 |
| **Performance Status ECOG** | | | | | | |
| 0 | Reference | Reference | | | | |
| 1 | 0.50 | (0.28–0.88) | 0.017 | | | |
| 2 | 0.81 | (0.40–1.62) | 0.549 | | | |
| 3 | 0.96 | (0.13–7.29) | 0.972 | | | |
| 4 | – | | | | | |
| **Smoking status** | | | | | | |
| Never | Reference | Reference | | Reference | Reference | |
| Current | 1.95 | (1.10–3.46) | **0.021**[*] | 1.27 | (0.69–2.35) | 0.438 |
| Former | 2.28 | (1.27–4.10) | **0.006**[**] | 1.45 | (0.76–2.75) | 0.261 |
| **Comorbidity** | | | | | | |
| CAD/HTN | 0.90 | (0.57–1.40) | 0.625 | | | |
| DM | 1.54 | (0.94–2.54) | 0.089 | | | |
| COPD/Asthema | 1.11 | (0.27–4.50) | 0.889 | | | |
| CVA | 1.45 | (0.53–3.97) | 0.468 | | | |
| CKD(Cr>1.5) | 1.35 | (0.83–2.20) | 0.223 | | | |
| Uremia at diagnosis | 1.30 | (0.70–2.41) | 0.399 | | | |
| HBV or HCVcarrier | 0.91 | (0.45–1.81) | 0.780 | | | |
| **Pathological T** | | | | | | |
| T0–1 | Reference | Reference | | Reference | Reference | |
| T2 | 2.17 | (0.99–4.74) | 0.052 | 2.07 | (0.89–4.81) | 0.091 |
| T3–4 | 6.17 | (3.57–10.68) | **<0.001**[**] | 4.42 | (2.20–8.88) | **<0.001**[**] |
| **Multifocalty** | | | | | | |
| No | Reference | Reference | | | | |
| Yes | 1.27 | (0.81–1.99) | 0.298 | | | |
| **Tumor cell differentiation** | | | | | | |
| CIS | | | | | | |
| Negative | Reference | Reference | | | | |
| Positive | 1.25 | (0.70–2.23) | 0.447 | | | |
| Tumor grading | | | | | | |
| G1 | Reference | Reference | | | | |
| G2–3 | 0.05 | (0.00–7.18) | 0.233 | | | |
| **Lymphovascular invasion** | | | | | | |
| Negative | Reference | Reference | | Reference | Reference | |
| Positive | 4.46 | (2.83–7.03) | **<0.001**[**] | 1.40 | (0.78–2.52) | 0.261 |
| **Surgical margin** | | | | | | |
| Negative | Reference | Reference | | Reference | Reference | |

(*Continued*)

**Table 2.** (Continued)

| | CSS | | | | | |
|---|---|---|---|---|---|---|
| Positive | 8.49 | (5.19–13.92) | **<0.001**** | 3.37 | (1.81–6.28) | **<0.001**** |
| **Adjuvant chemotherapy** | | | | | | |
| No | Reference | Reference | | Reference | Reference | |
| Yes | 2.09 | (1.30–3.34) | **0.002**** | 0.82 | (0.47–1.44) | 0.484 |
| | OS | | | | | |
| | Univariate | | | Multivariable | | |
| | Hazard ratio | 95%CI | *P*value | Hazard ratio | 95%CI | *P*value |
| **Gender** | | | | | | |
| Male | Reference | Reference | | | | |
| Female | 0.69 | (0.49–0.96) | **0.027*** | 0.90 | (0.54–1.49) | 0.679 |
| **Age, years** | 1.04 | (1.02–1.06) | **<0.001**** | 1.04 | (1.02–1.06) | **<0.001**** |
| **BMI** | 0.97 | (0.92–1.01) | 0.163 | | | |
| **Group** | | | | | | |
| PN0 | Reference | Reference | | Reference | Reference | |
| PN1–3 | 4.20 | (2.36–7.47) | **<0.001*** | 1.55 | (0.79–3.02) | 0.200 |
| PNx | 1.33 | (0.93–1.90) | 0.116 | 1.32 | (0.90–1.93) | 0.154 |
| **Performance Status ECOG** | | | | | | |
| 0 | Reference | Reference | | | | |
| 1 | 0.98 | (0.58–1.67) | 0.946 | | | |
| 2 | 1.72 | (0.95–3.12) | 0.074 | | | |
| 3 | 1.79 | (0.41–7.76) | 0.437 | | | |
| 4 | 3.46 | (0.46–26.12) | 0.229 | | | |
| **Smoking status** | | | | | | |
| Never | Reference | Reference | | Reference | Reference | |
| Current | 1.43 | (0.90–2.28) | 0.0134 | 0.98 | (0.53–1.81) | 0.947 |
| Former | 2.09 | (1.35–3.25) | **0.001**** | 1.59 | (0.87–2.90) | 0.135 |
| **Comorbidity** | | | | | | |
| CAD/HTN | 1.15 | (0.82–1.61) | 0.434 | | | |
| DM | 1.70 | (1.17–2.46) | **0.005*** | 1.57 | (1.06–2.34) | **0.025*** |
| COPD/Asthema | 1.29 | (0.48–3.49) | 0.618 | | | |
| CVA | 1.92 | (0.98–3.78) | 0.058 | | | |
| CKD(Cr>1.5, non-uremic status) | 1.38 | (0.96–2.00) | 0.084 | | | |
| Uremia at diagnosis | 1.62 | (1.05–2.49) | **0.030*** | 2.87 | (1.79–4.49) | **<0.001**** |
| HBV or HCVcarrier | 1.06 | (0.65–1.74) | 0.820 | | | |
| **Pathological T** | | | | | | |
| T0–1 | Reference | Reference | | Reference | Reference | |
| T2 | 1.56 | (0.91–2.67) | 0.104 | | | |
| T3–4 | 3.49 | (2.41–5.05) | **<0.001**** | 2.82 | (1.76–4.49) | **<0.001**** |
| **Multifocalty** | | | | | | |
| No | Reference | Reference | | | | |
| Yes | 1.31 | (0.93–1.84) | 0.117 | | | |
| **Tumor cell differentiation** | | | | | | |
| CIS | | | | | | |
| Negative | Reference | Reference | | | | |
| Positive | 1.05 | (0.66–1.67) | 0.834 | | | |
| Tumor grading | | | | | | |
| G1 | Reference | Reference | | | | |
| G2–3 | 0.05 | (0.00–1.89) | 0.105 | | | |

*(Continued)*

**Table 2.** (Continued)

| | CSS | | | | | |
|---|---|---|---|---|---|---|
| **Lymphovascular invasion** | | | | | | |
| Negative | Reference | Reference | | Reference | Reference | |
| Positive | 3.20 | (2.25–4.57) | **<0.001**** | 1.31 | (0.82–2.10) | 0.256 |
| **Surgical margin** | | | | | | |
| Negative | Reference | Reference | | Reference | Reference | |
| Positive | 4.77 | (3.05–7.45) | **<0.001**** | 2.75 | (1.61–4.72) | **<0.001**** |
| **Adjuvant chemotherapy** | | | | | | |
| No | Reference | Reference | | Reference | Reference | |
| Yes | 1.61 | (1.11–2.34) | **0.012**** | 1.04 | (0.66–1.65) | 0.866 |
| | DFS | | | | | |
| | Univariate | | | Multivariable | | |
| | Hazard ratio | 95%CI | *P*value | Hazard ratio | 95%CI | *P*value |
| **Gender** | | | | | | |
| Male | Reference | Reference | | | | |
| Female | 0.71 | (0.52–0.97) | **0.034*** | 0.90 | (0.55–1.46) | 0.665 |
| **Age, years** | 1.03 | (1.02–1.05) | **<0.001**** | 1.04 | (1.02–1.06) | **<0.001**** |
| **BMI** | 0.98 | (0.93–1.03) | 0.367 | | | |
| **Group** | | | | | | |
| PN0 | Reference | Reference | | Reference | Reference | |
| PN1–3 | 4.54 | (2.56–8.05) | **<0.001**** | 1.68 | (0.86–3.28) | 0.126 |
| PNx | 1.25 | (0.89–1.76) | 0.206 | 1.21 | (0.84–1.76) | 0.307 |
| **Performance Status ECOG** | | | | | | |
| 0 | Reference | Reference | | | | |
| 1 | 1.08 | (0.65–1.79) | 0.764 | | | |
| 2 | 1.71 | (0.96–3.06) | 0.068 | | | |
| 3 | 1.77 | (0.41–7.63) | 0.444 | | | |
| 4 | 3.58 | (0.48–26.92) | 0.215 | | | |
| **Smoking status** | | | | | | |
| Never | Reference | Reference | | Reference | Reference | |
| Current | 1.45 | (0.93–2.25) | 0.099 | 0.99 | (0.55–1.78) | 0.965 |
| Former | 1.89 | (1.23–2.92) | **0.004**** | 1.39 | (0.77–2.51) | 0.276 |
| **Comorbidity** | | | | | | |
| CAD/HTN | 1.08 | (0.78–1.50) | 0.627 | | | |
| DM | 1.66 | (1.16–2.38) | **0.006**** | 1.52 | (1.03–2.23) | **0.036*** |
| COPD/Asthema | 1.23 | (0.45–3.31) | 0.689 | | | |
| CVA | 1.99 | (1.05–3.78) | **0.036*** | 1.01 | (0.47–2.17) | 0.973 |
| CKD(Cr>1.5, non-uremic status) | 1.28 | (0.89–1.84) | 0.186 | | | |
| Uremia at diagnosis | 1.57 | (1.03–2.38) | **0.035*** | 2.60 | (1.62–4.17) | **<0.001**** |
| HBV or HCVcarrier | 1.06 | (0.66–1.73) | 0.798 | | | |
| **Pathological T** | | | | | | |
| T0–1 | Reference | Reference | | Reference | Reference | |
| T2 | 1.60 | (0.97–2.64) | 0.066 | | | |
| T3–4 | 3.18 | (2.24–4.53) | **<0.001**** | 2.37 | (1.46–3.85) | **<0.001**** |
| **Multifocalty** | | | | | | |
| No | Reference | Reference | | | | |
| Yes | 1.34 | (0.97–1.86) | 0.074 | | | |
| **Tumor cell differentiation** | | | | | | |
| CIS | | | | | | |

(*Continued*)

**Table 2.** (Continued)

| | CSS | | | | | |
|---|---|---|---|---|---|---|
| Negative | Reference | Reference | | | | |
| Positive | 1.14 | (0.74–1.76) | 0.556 | | | |
| Tumor grading | | | | | | |
| G1 | Reference | Reference | | | | |
| G2–3 | 0.15 | (0.02–1.04) | 0.055 | | | |
| **Lymphovascular invasion** | | | | | | |
| Negative | Reference | Reference | | Reference | Reference | |
| Positive | 3.18 | (2.26–4.47) | **<0.001**\*\* | 1.41 | (0.89–2.22) | 0.144 |
| **Surgical margin** | | | | | | |
| Negative | Reference | Reference | | Reference | Reference | |
| Positive | 4.77 | (3.08–7.38) | **<0.001**\*\* | 2.68 | (1.57–4.59) | **<0.001**\*\* |
| **Adjuvant chemotherapy** | | | | | | |
| No | Reference | Reference | | Reference | Reference | |
| Yes | 1.67 | (1.17–2.39) | **0.005**\*\* | 1.08 | (0.68–1.70) | 0.749 |

Cox proportional hazard regression.

\*$p < 0.05$

\*\*$p < 0.01$.

CSS = cancer-specific survival. OS = overall survival. DFS = disease-free survival. CI = confidence interval. ECOG = Eastern Cooperative Oncology Group. CAD/HTN = coronary artery disease / hypertension. DM = diabetes mellitus. COPD = chronic obstructive pulmonary disease. CVA = cerebrovascular accident. CKD = chronic kidney disease. HBV = hepatitis B virus. HCV = hepatitis C virus. CIS = carcinoma in situ. G = Grade.

[9 (3.2%) vs. 11 (24.4%), p<0.001]. The regional LND group also had more locally advanced diseases ($\geq$T2, 46.4% vs. 66.7%), p = 0.039) (Table 3).

## Discussion

This is a retrospective study to distinguish the prognostic impact of LND on patients treated with RNU with cN0 UTUC. As a result, LND for cN0 UTUC did not show therapeutic benefits in terms of DFS, CSS, and OS in the pN0 group. However, LND with RNU was observed to still lead a poor prognosis and allowed optimal tumor staging for further treatment if needed. Based on our study, the clinical T stage may indicate the need for LND. The regional LND should be performed on patients with cT3–4, or patients suspected to have sT3–4 disease during the operation, even though they had been considered cN0 at first. Moreover, for patients with characteristics of high grade tumors, positive lymphovascular invasion, or positive surgical margin, additional AC should be considered because of the risk of lymph node metastasis as implicated in our study.

Upper UTUC is a relatively rare disease with a prognosis poorer than that of bladder cancers [7–9]. The "gold standard" therapy for localized UTUC is RNU with ipsilateral BCE, but the role of LND in patients who are cN0 remains controversial [7, 13]. The presence of LN metastasis is associated with a poor prognosis [6, 24–26]. The reported incidence of LN metastasis was 37% for $\geq$pT3 disease, but only 3% for $\leq$pT2 disease [27]. The incidence of pN+ in patients with cN0 and $\geq$pT2 ranges from 14.3% to 40% [7]. Some studies have suggested performing LND at the time of RNU for patients with UTUC mainly due to the staging and therapeutic benefits of LND [22, 23]. In our present sample, 20 (3.8%) patients with pN+ disease had a significantly worse prognosis in terms of DFS, CSS, and OS compared with patients who were pN0. Our findings are consistent with the current literature.

**Table 3.  Association between types of LND and clinic-pathological characteristics of cN0 UTUC patients treated with RNU and BCE.**

|  | Hilar-only LND | Regional LND | *P* value |
|---|---|---|---|
| **Total case** | 278(86.1%) | 45(13.9%) |  |
| **number of dissected nodes†** | 1.0(0.0–5.0) | 11.0(6.0–41.0) | **<0.001**** |
| **Positive node** | 9(3.2%) | 11(24.4%) | **<0.001**** |
| **Pathological T** |  |  | **0.039*** |
| T0–1 | 149(53.6%) | 15(33.3%) |  |
| T2 | 35(12.6%) | 9(20.0%) |  |
| T3–4 | 94(33.8%) | 21(46.7%) |  |
| **Tumor grading** |  |  | 0.145 |
| G1 | 7(2.5%) | 0(0.0%) |  |
| G2 | 80(28.8%) | 8(17.8%) |  |
| G3 | 191(68.7%) | 37(82.2%) |  |
| **Hydronephrosis** | 22(7.9%) | 5(11.1%) | 0.559 |
| **Multifocal disease** | 107(38.5%) | 20(44.4%) | 0.552 |

Chi-square test.

†Mann-Whitney U test, Median (Range)

*$P<0.05$

**$P<0.01$

G = Grade.

According to the NCCN guidelines, patients with pT2–4 and pN+ UTUC should consider postoperative AC [13]. Although some observational studies have reported inconsistent results regarding the effectiveness of AC [28–32], a recent systematic review showed that AC is associated with better metastasis-free survival (HR = 0.65, p<0.001) and CSS (HR = 0.66, p<0.001). The association between AC and OS is significant in patients with locally advanced UTUC [33]. Seisen et al. found OS benefits of AC after RNU for patients with pT3/T4 and/or pN + UTUC [34]. The phase III POUT trial has demonstrated the benefit of adjuvant platinum-based chemotherapy for patients with locally advanced UTUC, in that AC significantly improved DFS (HR = 0.45, 95% CI = 0.30–0.68; p = 0·0001) [18]. The post-operative nodal status allows the selection of those patients (pN+) who may benefit from adjuvant systemic therapy. In our study, 70.0% of the patients in the PN+ group received AC, compared with 18.8% and 18.3% in the PN0 and PNx groups, respectively. Due to poor performance status, the others (30%) in the PN+ group did not receive AC. The PN+ group had a large proportion of a more aggressive tumor grade (G3, 100%), lymphovascular invasion (75%), and pathological T stage (>T2, 80%). However, there was no significant difference in terms of CSS, OS, and DFS when these were correlated with AC after balancing the confounders in the multivariate analysis for the entire population (HR = 0.82, CI = 0.47–1.44, p = 0.484, HR = 1.04, CI = 0.66–1.65, p = 0.866 and HR = 1.08, CI = 0.68–1.70, p = 0.749, respectively).

In contrast to the role of LND for UTUC staging, the therapeutic benefit of LND for UTUC remains controversial. Several studies have indicated that the prognosis of patients with pNx disease is poorer than that of those with pN0 disease, further demonstrating the therapeutic benefits of LND [35–37]. Abe et al. found significant differences in CSS among pN0, pNx, and pN+ patient groups. Notably, the survival difference between the pN0 and pNx groups remained significant in the multivariate analysis (HR = 3.36, 95% CI = 1.90–5.93, p<0.001) [35]. Similar results were found by Roscigno et al., in that pNx is significantly associated with a poorer prognosis (five-year CSS) than is pN0 in ≥pT2 populations (70% vs. 58% vs. 33%; p = 0.017 and p<0.01, respectively) [36, 37]. In our study, no difference was found between

the pNx and pN0 groups in terms of DFS, CSS, and OS. However there might be a worse trend in the pNx group than in the pN0 group in terms of five-year DFS and OS, indicating that the LND might have a potential survival benefit for the patient's survival. However, pN1–3 disease was also not an independent prognostic factor of CCS, OS, and DFS in our multivariate analysis (HR = 2.10, p = 0.049; HR = 1.67, p = 0.149 and HR = 1.68, p = 0.126, respectively), a finding that is consistent with other scales. These discrepancies in our results may have been due to few patient numbers (3.8%, 20/520) and the large proportion of advanced disease (T3–4, 80%, 16/20) in our pN+ group.

Most large scale studies found no therapeutic benefit of LND in the overall population [36–44]. In those studies, no statistically significant difference existed between the two groups in the overall population, but a clear benefit of LND was revealed when the focus was on patients with muscle-invasive or locally advanced UTUC. These patients had significantly better survival rates when compared with those in the pNx group [36–39]. LND benefits are less clear in patients who are cN0, just as how no benefit was found in our present study. This could have been due to selection bias, as LND was performed on those with more severe diseases.

In our subgroup analysis, advanced disease (pT3 and pT4) showed significantly more node metastases [T3–4 vs. T0–2: 13.91% (16/115), 1.92% (4/208), p<0.001]. The number of dissected nodes [1.0 (0.0–5.0) vs. 11.0 (6.0–41.0), p<0.001] and the rate of positive node metastasis [9 (3.2%) vs. 11 (24.4%), p<0.001] were significantly higher in the regional LND group. These patients with regional LND also had more locally advanced diseases (≥T2, 46.4% vs. 66.7%, p = 0.039) and showed a trend of having higher grade tumors and more advanced pathological T stages. A reasonable explanation is related to the personal preferences of our surgeons.

The limitations of the present study were its retrospective design and setting in a single center. The incidence of UTUC in Taiwan at the time of writing is higher than in other regions, which might affect the results of the analysis [45]. Due to the different strategy of LND that resulted from clinical circumstances, surgeons' personal preferences and evolution of surgical techniques, there may have been potential selection biases. In addition, there were 86 patients with muscle-invasive UTUC who did not undergo LND in the PNx group (86/197, 43.65%), and who were considered to be at a high potential risk for LN metastasis, which may have affected their survival. Only 20 patients were found to have pathologically confirmed lymph node metastases, which was a small sample size. Furthermore, there may have been additional unmeasured factors that we did not consider which may have affected the results, although multiple clinical variables were included in our study. Despite these limitations, the strengths of our study were its setting in a tertiary referral center and a large number of UTUC cases, most of which underwent lymphadenectomy (62.1%).

## Conclusions

LNDs for cN0 UTUC showed no therapeutic benefits in terms of DFS, CSS, and OS in the pN0 group. However, LND with RNU allowed optimal tumor staging. We found that 6.2% (20 pN +/323 with LND) of our patients had clinically proven occult LN metastasis. Furthermore, the clinical T stage may indicate the need for LND. Regional LND should be performed on patients with cT3–4 or patients suspected of sT3–4 disease during the surgical operation. Further prospective and well-controlled clinical trials should be done to better establish the impact of LND on patients with cN0 UTUC.

## Supporting information

**S1 Table. The DFS for three patient groups.** Kaplan-Meier analysis of DFS for 520 patients with pathologically proved lymph node status (pN1–3 and pN0) or without LND (pNx) in cN0

UTUC undergoing RNU with BCE.
(TIF)

**S2 Table. The CSS for three patient groups.** Kaplan-Meier analysis of CSS for 520 patients with pathologically proved lymph node status (pN1–3 and pN0) or without LND (pNx) in cN0 UTUC undergoing RNU with BCE.
(TIF)

**S3 Table. The OS for three patient groups.** Kaplan-Meier analysis of OS for 520 patients with pathologically proved lymph node status (pN1–3 and pN0) or without LND (pNx) in cN0 UTUC undergoing RNU with BCE.
(TIF)

## Author Contributions

**Conceptualization:** Jian-Ri Li, Kun-Yuan Chiu.

**Data curation:** Chuan-Shu Chen, Cheng-Kuang Yang, Shian-Shiang Wang, Chia-Yen Lin.

**Formal analysis:** Chia-Yen Lin.

**Methodology:** Chuan-Shu Chen, Jian-Ri Li, Chen-Li Cheng.

**Writing – review & editing:** Hsiang-Chen Hsieh, Chun-Li Wang, Chia-Yen Lin, Kun-Yuan Chiu.

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
