## [Decision Letter · Decision Letter 0]

5 Oct 2022

PONE-D-22-24467The prognostic impact of lymph node dissection for upper urinary tract urothelial carcinoma in patients with clinically node-negative diseasePLOS ONE

Dear Dr. Hsieh,

Thank you for submitting your manuscript to PLOS ONE. After careful consideration, we feel that it has merit but does not fully meet PLOS ONE’s publication criteria as it currently stands. Therefore, we invite you to submit a revised version of the manuscript that addresses the points raised during the review process.

We look forward to receiving your revised manuscript.

Kind regards,

Alessandro Rizzo

Academic Editor

PLOS ONE

**Comments to the Author**

1. Is the manuscript technically sound, and do the data support the conclusions?

Reviewer #1: Partly

Reviewer #2: Yes

Reviewer #3: Yes

2. Has the statistical analysis been performed appropriately and rigorously? 

Reviewer #1: Yes

Reviewer #2: N/A

Reviewer #3: I Don't Know

3. Have the authors made all data underlying the findings in their manuscript fully available?

Reviewer #1: Yes

Reviewer #2: Yes

Reviewer #3: Yes

4. Is the manuscript presented in an intelligible fashion and written in standard English?

Reviewer #1: No

Reviewer #2: No

Reviewer #3: No

5. Review Comments to the Author

Reviewer #1: Dear Editor, thank you so much for inviting me to revise this manuscript about UTUC.

This study addresses a current topic.

The manuscript is quite well written and organized. English should be improved.

Figures and tables are comprehensive and clear.

The introduction explains in a clear and coherent manner the background of this study.

We suggest the following modifications:

• Introduction section: although the authors correctly included important papers in this setting, we believe the systemic treatment scenario for UTUC patients should be better discussed in the introduction section and some recently published papers added (PMID: 35858936; PMID: 34387596; PMID: 33516645 ), only for a matter of consistency. We think it might be useful to introduce the topic of this interesting study.

• Methods and Statistical Analysis: nothing to add.

• Discussion section: Very interesting and timely discussion. Of note, the authors should expand the Discussion section, including a more personal perspective to reflect on. For example, they could answer the following questions – in order to facilitate the understanding of this complex topic to readers: what potential does this study hold? What are the knowledge gaps and how do researchers tackle them? How do you see this area unfolding in the next 5 years? We think it would be extremely interesting for the readers.

However, we think the authors should be acknowledged for their work. In fact, they correctly addressed an important topic, the methods sound good and their discussion is well balanced.

One additional little flaw: the authors could better explain the limitations of their work, in the last part of the Discussion.

We believe this article is suitable for publication in the journal although major revisions are needed. The main strengths of this paper are that it addresses an interesting and very timely question and provides a clear answer, with some limitations.

We suggest a linguistic revision and the addition of some references for a matter of consistency. Moreover, the authors should better clarify some points.

Reviewer #2: Title:

- It is better to change it to "The prognostic impact of lymph node dissection for upper urinary tract urothelial carcinoma in patients treated with radical nephroureterectomy with clinically node-negative disease. General:

- There are some grammatical errors in the manuscript. Please improve the language quality of your revised manuscript.

Abstract: Conclusion:

- " The LND still had a prediction of a poor prognosis" it should be mentioned in this section.

Materials and Methods:

- The recorded dissected regions and the extent of LND should be clearly mentioned.

- Adjuvant chemotherapy should be mentioned.

Result:

- this section needs revision. I suggest seeing the previously published article with DOI: 10.1093/jjco/hyx051.

- It is better to report the mean follow-up duration with months.

- Adjuvant chemotherapy results, number of causes of mortality, the estimated OS at 2 and 5 years, and disease recurrence should be mentioned.

- Tables: all abbreviations should be mentioned below the tables.

- Table 2 was mentioned twice; please revise it.

- Table 2 shows the overall CSS and OS. However, the study was conducted to compare three groups. I think; this table should be revised.

Discussion:

- For studies, your Discussion section should first reiterate briefly the results, then move to a discussion of your main findings, and finally move to wider topics and comparison of your study with ,other research.

- The reported 5-year survival, mortality, and recurrence should be discussed.

- Add more limitations of this study. I suggest seeing the previously published article with DOI: 10.1093/jjco/hyx051.

References - Journal names should be abbreviated as per the Journals Database section in PubMed (http://www.ncbi.nlm.nih.gov/nlmcatalog/journals).

Reviewer #3: A single-center retrospective analyses of patients with UTUC + cN0 disease who underwent RNU +/- LND.

My comments are listed as below:

1. The manuscript needs some editing for various grammar and punctuation errors.

2. Some of the results have been written first in the Materials & Methods section, and then repeated in the Results section (e.g., the first and the third paragraphs of the M&M section). I recommend the authors to give all results in the Results section, which will facilitate its reading.

3. There is no data about how many surgeons performed the surgeries and what were their levels of expertise.

4. As a general rule the average (mean) value is given with standard deviation (SD), and the median is given with range or IQR. The FU time was given as mean with range. I recommend the authors to use a consistent method to give all results.

5. I think the authors wanted to mean lymphovascular invasion (LVI) with "angiolymphatic permeation" in the first paragraph of the Results section. Please make it consistent with the remaining part of the manuscript, such as table 1.

6. Please shorten the legend of the tables by using appropriate abbreviations.

7. Table 2 was written twice. It would be better to give all tables at the end of the manuscript.

8. The Results section would be re-written as they repeat all the information given in the tables. ı recommend the authors to summarize the most important results in this section and give concise information to th readers so that they can easily read it.

9. The authors have not mentioned about the POUT trial in the Discussion section where they give information about the use of NAC/AC in UTUC management.

10. I also recommend the authors to shorten the general information about UTUC and its management in the beginning of the Discussion section. Preferably, they would start directly with their findings and then compare them with that of previously published papers. If they want to give some information about UTUC here, it would be different from that of given in the Introduction section.

11. I do not agree with the authors that LND should be considered when the characteristics of high-grade tumor and LVI are detected in the patients. Do the authors mean that we should take these parameters into account from the pathology report of URS+biopsy? During RNU, we cannot see the tumor and we cannot have an impression that it is high grade. Please clarify this.

12. For me it is not clear why some cN0 patients have received LND while the others have not. What were the selection criteria? I assume that there would be some selection biases (as it is a retrospective trial). Please give data for this (maybe a separate table comparing the preoperative data of the cN0 patients who underwent and did not undergo LND).

13. I have not seen the result of 6.2% rate of occult LN metastasis in the Results section, however, this is written in the Conclusion parts of the text and the abstract. Please clarify this.

6. PLOS authors have the option to publish the peer review history of their article (what does this mean?). If published, this will include your full peer review and any attached files.

Reviewer #1: No

Reviewer #2: **Yes: **Faisal Ahmed

Reviewer #3: **Yes: **Murat AKAND

Please note that if a reviewer has requested citations to specific articles, those articles should only be cited if they are directly relevant to the study. If you find that any number of the requested citations are irrelevant or inappropriate, please state this in your Response to Reviewers for each article you assess to be irrelevant or inappropriate to cite.

---

## [Author Response · Author response to Decision Letter 0]

7 Nov 2022

Dear Reviewers

Thanks so much for revising my manuscript. The following are individual responses to the reviewers.

To: Reviewer #1

1. Introduction section : we added systemic treatment scenario for UTUC with the recently published papers you suggested (PMID: 34387596; PMID: 33516645 )

2. Discussion section: we summarized and expand our discussion 

The potential of our study may indicate that LND with RNU still had a prediction of a poor prognosis and allow optimal tumor staging for further treatment.

Further prospective and well‐controlled clinical trials should be done to better establish the impact of lymph node dissection for UTUC in patients with CN0 disease.

3. We added more limitations of this study

4. We had arranged a linguistic revision this time

To: Reviewer #2

1. Title: The title changed and modified to " The prognostic impact of lymph node dissection for clinically node-negative upper urinary tract urothelial carcinoma in patients who are treated with radical nephroureterectomy" as your suggestion.

2. General: We tried to improve the language quality of our revised manuscript this time

3. Abstract: Conclusion: "patients still had a poor prognosis" was mentioned in this section as your suggestion.

4. Materials and Methods:

The recorded dissected regions and the extent of LND and adjuvant chemotherapy were mentioned in this section.

5. Result: we had revised this section as your suggestion

We used mean follow-up duration with months and SD.

Adjuvant chemotherapy results, number of causes of mortality, the estimated OS at 2 and 5 years, and disease recurrence were mentioned in results or tables.

6. Tables: 

All abbreviations were mentioned below the tables.

Table 2 was revised, and included CSS, OS, and DFS (the data also corrected)

7. Discussion:

We summarized and expand our discussion as your suggestion.

The CSS, OS, and DFS were discussed.

Add more limitations of our study.

To: Reviewer #3

1. We had arranged a linguistic revision this time

2. All results were given in the Results section as your suggestion (removed from Materials & Methods section)

3. We provided the data about how many surgeons performed the surgeries and what were their levels of expertise.

4. We used mean follow-up duration with months and SD

5. We made it consistent of lymphovascular invasion (LVI) as your suggestion.

6. Table 2 was revised, and included CSS, OS and DFS.

7. We added POUT trial in the Discussion section as your suggestion about the use of NAC/AC in UTUC management.

8. We started directly with our findings and then compared them with that of previously published papers in discussion section as your suggestion.

9. “LND should be considered when the characteristics of high-grade tumor and LVI are detected in the patients. Do the authors mean that we should take these parameters into account from the pathology report of URS+biopsy? During RNU, we cannot see the tumor and we cannot have an impression that it is high grade. Please clarify this.”

It’s was our mistake and this paragraph should be “for patients with characteristics of high grade tumor, positive lymphovascular invasion or positive surgical margin, additional AC should be considered because of the risk of lymph node metastasis as implicated in our study”, not “LND”. Thanks for your correction.

10. The selection criteria of LND for cN0 patients was mentioned in limitation

Due to the different strategy of LND that resulted from clinical circumstances, surgeons’ personal preferences and evolution of surgical techniques, there may have been potential selection biases.

11. “I have not seen the result of 6.2% rate of occult LN metastasis in the Results section, however, this is written in the Conclusion parts of the text and the abstract. Please clarify this.”

The 6.2% (20 patients with pN+/ 323 patients who underwent LND) of our patients with clinically proven occult LN metastasis. Sorry about our unclear description.

We look forward to hearing good news about our revised manuscript.

Kind regards, 

Hsiang-Chen Hsieh, MD.

Taichung Veteran General Hospital, Taiwan

Department of Surgery, Division of Urology

Phone: +886-975-960069

E-mail: stilllove3q@hotmail.com

Chia-Yen Lin, MD (Corresponding Author)

Taichung Veteran General Hospital, Taiwan

Department of Surgery, Division of Urology

Mail: lcyhank.tw@gmail.com

---

## [Editor Report · Decision Letter 1]

9 Nov 2022

The prognostic impact of lymph node dissection for clinically node-negative upper urinary tract urothelial carcinoma in patients who are treated with radical nephroureterectomy

PONE-D-22-24467R1

Dear Dr. Lin,

We’re pleased to inform you that your manuscript has been judged scientifically suitable for publication and will be formally accepted for publication once it meets all outstanding technical requirements.

Kind regards,

Alessandro Rizzo

Academic Editor

PLOS ONE
---

## [Editor Report · Acceptance letter]

21 Nov 2022

PONE-D-22-24467R1 

The prognostic impact of lymph node dissection for clinically node-negative upper urinary tract urothelial carcinoma in patients who are treated with radical nephroureterectomy 

Dear Dr. Lin:

I'm pleased to inform you that your manuscript has been deemed suitable for publication in PLOS ONE. Congratulations! Your manuscript is now with our production department. 

Kind regards, 

on behalf of

Dr. Alessandro Rizzo 

Academic Editor

PLOS ONE